# SOM-LWL method for identification of COVID-19 on chest X-rays

**Ahmed Hamza Osman** [1]*, **Hani Moetque Aljahdali** [1], **Sultan Menwer Altarrazi**[2], **Ali Ahmed** [2]

**1** Department of Information System, Faculty of Computing and Information Technology, King Abdulaziz University, Rabigh, Saudi Arabia, **2** Department of Computer Science, Faculty of Computing and Information Technology, King Abdulaziz University, Rabigh, Saudi Arabia

* ahoahmad@kau.edu.sa

**Data Availability Statement:** All the image files are available from GitHub (https://github.com/muhammedtalo/COVID-19).

**Funding:** This project was funded by the Deanship of Scientific Research (DSR) at King Abdulaziz

## Abstract

The outbreak of coronavirus disease 2019 (COVID-19) has had an immense impact on world health and daily life in many countries. Sturdy observing of the initial site of infection in patients is crucial to gain control in the struggle with COVID-19. The early automated detection of the recent coronavirus disease (COVID-19) will help to limit its dissemination worldwide. Many initial studies have focused on the identification of the genetic material of coronavirus and have a poor detection rate for long-term surgery. The first imaging procedure that played an important role in COVID-19 treatment was the chest X-ray. Radiological imaging is often used as a method that emphasizes the performance of chest X-rays. Recent findings indicate the presence of COVID-19 in patients with irregular findings on chest X-rays. There are many reports on this topic that include machine learning strategies for the identification of COVID-19 using chest X-rays. Other current studies have used non-public datasets and complex artificial intelligence (AI) systems. In our research, we suggested a new COVID-19 identification technique based on the locality-weighted learning and self-organization map (LWL-SOM) strategy for detecting and capturing COVID-19 cases. We first grouped images from chest X-ray datasets based on their similar features in different clusters using the SOM strategy in order to discriminate between the COVID-19 and non-COVID-19 cases. Then, we built our intelligent learning model based on the LWL algorithm to diagnose and detect COVID-19 cases. The proposed SOM-LWL model improved the correlation coefficient performance results between the Covid19, no-finding, and pneumonia cases; pneumonia and no-finding cases; Covid19 and pneumonia cases; and Covid19 and no-finding cases from 0.9613 to 0.9788, 0.6113 to 1 0.8783 to 0.9999, and 0.8894 to 1, respectively. The proposed LWL-SOM had better results for discriminating COVID-19 and non-COVID-19 patients than the current machine learning-based solutions using AI evaluation measures.

## 1. Introduction

A new disease that occurred in 2019 and that was not known previously in humans was coronavirus disease 2019 (COVID-19). Coronaviruses (CoVs) are a wide variety of viruses that

University, Jeddah, Saudi Arabia under grant No.
(GCV19-6-1441). The authors, therefore, gratefully
acknowledge DSR for technical and financial
support.

**Competing interests:** The authors have declared
that no competing interests exist.

cause respiratory diseases such as severe acute respiratory syndrome (SARS-CoV) and Middle
East respiratory syndrome (MERS-CoV). The new coronavirus started to spread in China in
December 2019 and later spread to many other countries [1–3]. It be very helpful to reduce the
spread of this disease by early automated diagnosis [4]. Deep learning is one of the most help-
ful methods of artificial intelligence for detecting COVID-19 infections from medical images,
such as X-rays, especially when a limited image dataset is accessible [2, 4]. Past experiments
have used deep learning from chest X-rays for the identification of COVID-19. By including
71 COVID-19 and 5000 non-COVID 19 images, Minaee et al. [2] evaluated a dataset of 5071
chest X-rays. They selected 40 COVID-19 and 3000 COVID-19 cases not included in the sur-
vey plus 31 COVID-19 (496 post increase) and 2000 COVID-19 cases not included in the
training set images. They trained 4 common deep learning models to detect COVID-19 infec-
tions, such as ResNet18, ResNet50, DenseNet-121 and SqueezeNet. At 97.5% accuracy, the
best-performing model reached 95% specificity [5]. The output of numerous state-of-the-art
CNN archives of two datasets was analysed by Apostolopoulos and Mpesiana [4]. The first
contained 1427 chest X-rays, 224 COVID-19 images, seven hundred confirmed common bac-
terial pneumonia images and 504 images from healthy patients. The second dataset contained
224 COVID-19 images, 714 bacterial and viral pneumonia verified images and 504 stable case
images [4]. Their findings indicated that the highest accuracy, sensitivity and specificity were
96.78%, 96.46%, and 98.66%, respectively [4]. The DarkCovidNet Model for detecting
COVID-19 from chest X-rays was proposed by Ozturk et.al [6]. Our model was tested on a
dataset containing 125 COVID-19 chest X-rays and 500 balanced chest X-rays. Their model
provided an accuracy of diagnosis of 98.08% (Healthy vs COVID-19) and 87,02% (Healthy vs
COVID-19 vs. Pneumonia) for non-binary classification cases [6]. COVID-19 diagnosis is
normally related to all the symptoms in the chest X-ray scans for pneumonia [7]. The first
screening procedure is a chest X-ray. It plays a major role in COVID-19 diagnosis. COVID-19
treatment is usually related both to pneumonia symptoms and to chest rays [7]. Chest X-rays
have become the first imaging tool to play a significant role. Recently, widely accessible X-rays
have not only improved in stable cases but also improved in patients with COVID-19. This
helps one to research diagnostic images and recognize potential variations that will result in
the illness being immediately identified. In chest X-rays, patients had peculiar conditions. The
disease's symptoms cause breathing issues, cardiac injury, and secondary infection. The results
revealed that COVID-19 spreads independently. The infected person must be treated in the
intensive care unit when severe respiratory problems occur. Radiography of affected individu-
als reveals unique features of COVID-19. Therefore, clinical experts need chest X-ray images
for early diagnosis of COVID-19. Chest X-ray studies have found that the COVID-19-related
lunar shadowing sensitivity has been reduced by 25% to 69% [8, 9]. On the other hand, the
specificity of this technique for properly identifying the disease is 90% [8]. The research veri-
fied the use of RT-PCR in all cases of COVID-19. The small number of participants (17 in [8]
and 64 in [9]) could have contributed to the discrepancies in sensitivity. The time between ini-
tial symptoms and the imaging procedure may be a significant factor affecting the reliability of
X-ray findings. Although in the first 3 days after the onset of coughing and fever the symptoms
are not yet apparent on X-rays, they are the most visible after 10–12 days. This time factor
appears to be supported by an Italian analysis of 72 symptomatic patients released in mid-
April 2020. All patients had already been under quarantine at home and were hospitalized
because their symptoms worsened when the imaging procedure was carried out. The sensitiv-
ity of chest X-rays was 69% (no information about the specificity was provided) [8]. While
there are comparatively limited numbers of cases covered by individual X-rays of COVID-19
patients, a collection of characteristic results [6, 8, 9] shows that the most common lung shifts
are concentration of fluid and/or tissue in the lungs preventing gas exchange in the pulmonary

alveoli. In addition, the ground glass opacities and shadowing nodules primarily affect the peripheral and lower lung regions. In view of the uncertain data situation, medical societies and professional bodies have aimed to provide advice. In view of this data situation, the focus is on CT. The Radiological Society of North America (RSNA) provided an expert consensus statement that states that CT is not currently recommended for screening to diagnose or exclude COVID-19 [10]. The Fleischer Company confirmed in its statement that chest X-rays are insensitive in the early stages of the disease. Nonetheless, X-ray analysis shows regular lung changes as quarantined patients with severe symptoms are studied. Chest X-rays may be adequate to evaluate the course of the disease and evaluate pneumonia for other reasons, according to the Fleischer Society's view [11], in patients who are already hospitalized. Therefore, for COVID-19 patients in intensive care who are not sufficiently healthy to undergo CT scans, the European Society of Thoracic Imaging (ESTI) and European Society of Radiology (ESR) recommend the use of X-ray imaging [12].

The main advantages and contributions of this research are:

1. Suggested a COVID-19 prediction method can improved the diagnosis accuracy and decrease the miss diagnosis error when integrated some supervised and unsupervised machine learning techniques.

2. The main advantages and contributions of this research is that the locality-weighted learning algorithm has been adapted by adding a clustering process to the dataset before using the LWL, which we call the SOM-LWL model for the identification of COVID-19 cases from chest X-ray findings.

3. The SOM clustering method has been applied to pre-trained models with single and multi-class datasets. The clustering process aims to split the dataset samples and classes into many subsamples and subclasses within the image dataset and then assign new clustering labels to the new set, under which each subject set is viewed as a separate class.

4. The similarity and diversity of these clusters is highlighted in the dataset instances, consequently helping to identify variations among members of the dataset and facilitating the classification and learning process when constructing the LWL diagnostic model.

5. Radiological imaging method is used to emphasizes the performance of chest X-rays with different type of cases such as positive COVID-19, Non-COVID-19, and pneumonia cases.

On the contrary, the limitation of the proposed method is that it focuses only on chest x-rays dataset, while there are other medical datasets can be used to detect the COVID-19.

The following are the other parts of this paper: Section 2 addresses the related research of this study. The descriptions of the proposed SOM-LWL scheme are provided in Section 3. The approach and methodology are explained in Section 4. Section 5 provides descriptions of the experimental findings and dataset. Section 6 provides a description of the results, discussion and analysis. Section 7 is a summary and discusses future works related to the study.

## 2. Related work

Real-time reverse transcription-polymerase chain reaction (RT-PCR) is the primary research technique currently in use for COVID-19 diagnosis. Chest radiographic images, such as CT images and X-rays, are critical for the early diagnosis and treatment of the condition [10]. The low sensitivity of RT-PCR (60–70%) allows symptoms to be detected by analysing radiographic images of patients, even though adverse findings are obtained [11, 12]. CT is a sensitive diagnostic tool for COVID-19 pneumonia diagnosis and can be used as an RT-PCR screening tool

[13]. CT results are often found long after symptoms occur, and patients typically undergo CT analysis within the first 0 to 2 days [12]. In research on lung X-rays, the most severe lung illness was found 10 days after symptoms were shown in patients who survived COVID-19 pneumonia [14]. During the onset of the pandemic in China, inadequate diagnostic kits were available at health centres, and high levels of false negative tests were reported, such that doctors were advised to use health examinations and chest CT scans for diagnosis [15, 16]. CT has been used in countries such as Turkey, where a small number of test kits were available at the onset of the pandemic, for COVID-19 diagnosis. Researchers have suggested that comparing clinical imaging findings with laboratory tests will help to diagnose COVID-19 early [7, 13, 17, 18]. The diagnostic information in radiographic images collected from COVID-19 patients is valuable. Several reports have indicated improvements before the effects of COVID-19 began based on chest X-rays and CT scans [19]. Researchers have also made important advances in COVID-19 imaging research. In a COVID-19 case, Kong et al. [17] noted right-sided ground glass opacity. Yoon et al. [20] found a single nodular opacity in the lower left lung area in one out of three examined patients. The other two patients, by comparison, displayed abnormal hardness between lung areas four and five. Zhao et al. [21] noted a convergence and vascular dilation in the lesions in multiple patients as well as mixed GGOs. As typical CT features of COVID-19 patients, Li and Xia [18] reported GGOs and condensed air, interlobular septal thickening, and indications of bronchograms with or without vascular expansion. Another finding was that lateral foci or multifocal GGOs in both lungs affect 50% to 75% of patients [11]. Likewise, Zu et al. [10] found that rounded lung illumination can be identified in 33% of chest CT scans.

Rasheed et al. [22] introduced a survey paper investigated medical and technical viewpoints in the battle against the epidemic of COVID-19, which will support virologists, IA researchers and policymakers. The paper also discussed and understood the usage of various technical instruments and techniques within COVID-19. In addition, the study reveals a variety of AI approaches proposed to support the COVID-19 pandemic, from initial diagnosis through image diagnostics via models which help to explain COVID-19 spread and recognize new possible spread areas for the outbreak. The use of predictive diagnostic machine learning approaches has recently gained attention in the medical industry as a critical resource for clinicians [23–28]. Deep learning, a common field of artificial intelligence (AI), allows the creation of models end-to-end in order without requiring manual feature extraction to produce predicted results using input data. Several approaches have been proposed a deep learning methods for the identification of COVID-19 events such as CNN [29–31], COVIDScreen [32], and COVINet [32]. These approaches were used an efficient and robust X-ray and CT scan imaging solutions.

A variety of problems such as identification of arrhythmias [33], diagnosis of skin cancer [34], identification of breast cancer [28, 35], surgical diagnosis [36], identification of pneumonia [37], segmentation of the fundus [38] and lung segmentation [39] have been evaluated effectively by deep learning techniques. The rapid spread of the COVID-19 outbreak has demanded expertise. The development of automatic detection systems based on AI techniques has increased in interest. Because of the small number of radiologists, this technology is a daunting challenge for specialist clinics at any hospital. Therefore, it can be useful to solve this problem by supplying patients with quick, precise, and fast AI models. While radiologists play an important role in achieving an accurate diagnosis due to their extensive expertise in the field, AI technology can also be used in radiology [40]. Furthermore, AI procedures can help to eliminate drawbacks such as an insufficient number of usable RT-PCR test kits and test costs. Recently, Sedik, A et al. [41] improved the learning capacities of the Convolutional Neural Network (CNN) and CLSTM-based deep learning models (DADLMs)

by introduced a two machine learning models to in order to enhance the prediction accuracy of COVID-19 identification. Several radiographic images for the identification of COVD-19 were commonly used. To diagnose COVID-19 in X-rays, Hemdan et al. [42] used deep learning algorithms, suggesting a COVIDX-Net network containing seven CNN models. The deep learning COVID 19 (COVID Net) model, which had an accuracy of 92.4%, was suggested by Wang and Wong [43] to define groups as regular, non-COVID, and COVID-19. Using 224 confirmed COVID-19 images, Ioannis et al. [44] established a deep learning pattern. Their model achieved success rates of 98.75% and 93.48% for all three levels. A 98% COVID-19 detection by chest X-ray signal, along with the ResNet50 pattern, was obtained by Narin et al. [4]. Similarly, Haque, K.F. and Abdelgawad, A [45] proposed a CNN model for detecting a COVID-19 positive patients. This model identifies Coronavirus patients with very little time and energy, and is very accurate. In their work, the CNN models in COVID-19 are also studied in a comparative analysis.

Sethy and Behera [46] have identified the features extracted from various CNN models using X-ray images and employed a support vector machine (SVM). Their analysis notes the highest results of the ResNet50 model with the SVM classifier. Finally, some recent COVID-19 experiments employed a variety of CT image deep learning models [47].

Recently, an algorithm based on laboratory and demographic features was proposed by Goodman-Meza D. et al. [48] to serve as a screening method in hospitals where testing is limited or inaccessible. The methodology used data obtained retrospectively from the UCLA Health System in Los Angeles, California. The study included all emergency area or inpatient cases that included SARS-CoV-2 PCR testing during March and May 2020, as well as a collection of ancillary laboratory features (n = 1,455).

Bird J.J. and Barnes CM A. et al. [49] proposed a three-step machine learning approach for country-level risk prediction based on disclosed COVID-19 data, and these data are used in this review. K-percent binary discretisation (K = 25) is used to establish four risk categories for countries based on the risk of infection (coronavirus cases per million people), the risk of death (coronavirus deaths per million people) and the risk of failure to test (coronavirus tests per million people). 'Low', 'medium-low', 'medium-moderate' and 'high' are the four risk groups created by K-percent binning. Coronavirus-related data are then deleted, and the characteristics of the three categories of risk prediction are given considering the geopolitical and demographic data describing each region. Via a cross-validation strategy with a leave-one-country-out technique, three four-class classification issues are then investigated and benchmarked to find the best model; SGB and DT algorithms are established for transmission danger, and extra tree and stack SVM algorithms are proposed for death and testing limitation risks.

Elaziz MA. et al. [50] Suggested a COVID-19 machine learning method to classify X-ray images of the chest into two groups: COVID-19 or non-COVID-19 patients. Their model used a FrMEMs method to exploit the features from chest X-ray images. To accelerate the computational process, a parallel multi-core computational architecture was used. Then, the most important features were selected using modified manta ray foraging optimization based on differential evolution.

In this research, an automated identification of COVID-19 is proposed in a hybrid unsupervised and supervised learning model represented by SOM-LWL. To bypass the treatment, the current model requires an end-to-end structure without using any extraction approaches. This sample consists of 125 images of chest X-rays that are not standard and have been obtained rapidly. More reliable diagnostic methods are therefore required. One of the most significant drawbacks of chest X-ray studies is the fact that they cannot detect early COVID-19 phases, since they are not adequately sensitive in GGO detection [10]. However, well-trained

deep learning models will reflect problems that are not apparent to the human eye and can change this perception. Table 1 shows the summary of the related work methods.

## 3. Proposed SOM-LWL model

Despite their self-learning capacity and superior prediction performance, LWL and SOM models achieve human-like precision in image description and prediction issues. Our framework aims mainly at providing distinguishing visual properties and a quick diagnostic system that can be used to classify new COVID-19 X-rays. This technique can also be useful to clinicians as a treatment plan that can be used depending on the type of infection and can provide prompt decisions. The following sections describe the suggested operational framework, design of the SOM-LWL scheme, and the solution of the imbalanced X-ray dataset. The operational framework is demonstrated in Fig 1.

Fig 1 presented the three phases of the general structure of the SOM-LWL based diagnostic scheme.

The suggested model is collected of three key phases: the imbalanced raw dataset and feature extraction, clustering of the data instances based on similarity of the patients features using the SOM model, and decision-making diagnosis with the training and testing phase using the LWL prediction model.

The suggested model categorizes the classes of X-rays labelled as Non-COVID (viral-infection), COVID (COVID-19 viral-infection), and pneumonia (microbial-infection).

### 3.1. Imbalance data handling

In the first phase, the imbalanced data have been handled by utilizing the raw input features of X-rays due to their irregular sample distributions. The method used to solve this problem is to divide all the dataset into equal parts for each class. For example, the number of confirmed cases for COVID-19 patients is 125 cases, while the number of cases for non-COVID-19 sufferers is 500, and those with pneumonia are 500 cases. The non-infected cases and the cases of pneumonia were divided into four parts of equal value, each part consisting of 125 samples, and the samples of each of these four groups equal the samples of the groups of people with COVID-19. We repeated the joining of the COVID-19 samples for each group separately and used this dataset as the crossover with other generated non-infected and pneumonia groups. The number of classes of generated groups was created equally such that each group contains 375 cases consisting of 125 patients infected with COVID-19 and labelled class 1, as well as 125 non-infected patients with COVID-19 and 125 cases of pneumonia that are labelled with class 2 and class 3, respectively. The four groups (A, B, C, and D) that are labelled from this process were included in the diagnosis clustering and classification experiments for each group individually. The imbalanced data handling process is demonstrated in Fig 2.

Fig 2 demonstrates the imbalanced data that have been handled by utilizing the raw input features of X-rays due to their irregular sample distributions.

### 3.2. Locally weighted learning method (LWL)

In a region around the query example, the locally weighted regression (LWR) attempts to modify the training data. LWR is a form of lazy learning, so training data are typically delayed until a query example's target value must be forecast. LWR and regression of the kernel [51] are analogous to data distributed from every boundary on a normal grid. However, in abnormal data distributions, LWR outperforms kernel regression [52]. LWR has the best convergence rates in the minimum sense [53]; among all possible estimators, it has high minimum efficiency [54, 55]. Hastie & Loader [56] also showed that a number of data distributions are

**Table 1. Summary of the related work methods.**

| Reference | Method | Performance | Advantages | Disadvantages |
|---|---|---|---|---|
| [10], [11] | Thin-slice chest CT | A full score for COVID-19 in 155 of the 167 patients (92.8%) | The low sensitivity of RT-PCR screening tools (60–70%) allows symptoms to be detected by analysing radiographic images of patients. Thin-slice chest CT is simple to administer, swift, and highly sensitive to early COVID-19 pneumonia, offering useful evidence for further diagnosis while helping to avoid and monitor COVID-19. | • The method used the CT tool which it is a sensitive diagnostic tool for COVID-19 pneumonia diagnosis.<br>• CT results are often found long after symptoms occur, and patients typically undergo CT analysis within the first 0 to 2 days |
| [14] | • Crazy-paving pattern and GGO.<br>• Quantitative analysis using SPSS. | A cumulative CT score of 0 (no involvement) to 25 (maximum involvement) was calculated as the amount of lung inference. | Determine improvements from original diagnostic up to patient recuperation with COVID-19-related Chest CT findings. | In research on lung X-rays, the most severe lung illness was found 10 days after symptoms were shown in patients who survived COVID-19 pneumonia |
| [13] | RT-PCR | 60–70% sensitivity | The low sensitivity of RT-PCR (60–70%) allows symptoms to be detected by analysing radiographic images of patients. but on initial negative RT-PCR | CT results are can be found in initial negative RT-PCR only due to abnormalities on chest CT scan images. |
| [10] | Thin-slice chest CT | A full dant for COVID-19 in 155 of the 167 patients (92.8%) | The CT system used for COVID-19 results involves multifocal floor-to-ground (GGO's) peripherally scattered with patchy consolidations and tastes in the back and under lobe. In early identification, observation and disease assessment, chest CT played a crucial role. | It is uncertain that if chest x-rays are regular, the criterion for undertaking CT tests of probable lung changes may be smaller. Further experiments are required to increase the selection of CT patients, to identify the effectiveness of CT in COVID-19 pneumonia and to investigate the use of artificial intelligence in chest X-rays in suspicious cases.<br>The COVID-19 can be detected only using CT data only rather than other types of dataset. |
| [20] | Radiographical and CT analyses | The performance test of Fisher was used to equate CT findings according to the type of pulmonary lesions. | The radiographical and CT analyses from baseline pneumonia COVID-19 have been examined. The exact test of Fisher was used to equate CT findings according to the type of pulmonary lesions. | In this analysis, there are a few limitations. As of February 16, 2020, nearly one-third of all 29 COVID-19 patients in Korea were included in the group of patients, which was a small number.<br>Secondly, the approach relies on the baseline CT observations, which doctors and radiologists found first rather than the outcomes from follow-up CT scans.<br>Third, the procedure reduced patients' health knowledge as the study culminated in a large percentage of the patients undergoing inpatient therapy. |
| [12] | Negative RT-PCR | The mean score was 6.8 and the score for the median CT attendance was 4 (maximum CT score, 14; minimum CT score, 2). | Chest CT confirmation of viral pneumonia can be preceded by positive reverse transcriptional reaction test results in patients at risk for COVID-19. | CT findings appear late after occur of symptoms and usually CT scans for patients within the first 0 to 2 days are done.<br>High dosage and cost scanning of patients are the principal downside of using CT imagery. |
| [16] | rRT-PCR | CT sensitivity at present was 97.2%, although the original RRT-PCR sensitivity was just 83.3%. | The method can evaluated the CT and rRT-PCR diagnostic significantly for pneumonia COVID-19. | The availability of nucleic acid detection kits was limited since there was a COVID-19 pneumonia epidemic. Only in fever-positive and CT-positive cases were tested rRT-PCR. Furthermore this analysis had a limited sample size and due to time limitations, no follow-up was done. Consequently, for further verification, greater sampling sizes are necessary. |

*(Continued)*

**Table 1.** (Continued)

| Reference | Method | Performance | Advantages | Disadvantages |
|---|---|---|---|---|
| [17] | Viral pneumonia CT diagnosis Method | In chest CT there was a low diagnosis incidence of COVID-19 missing (3.9%, 2/51) | The method can able to determine and evaluate the mis-diagnosis error of radiologists for COVID-19 | The method still limited for recognising distinguish viruses and distinctive between them. During the research time the number of patients was reduced by the lack of laboratory test kits. |
| [21] | Chest CT Interaction Results and Coronavirus Clinical Conditions | In emergency patients, the prevalence of diffuse lesions was higher than in the non-emergency population (78.6% vs 24.1%). | The study discusses medical and technical viewpoints to promote the outbreak of COVID-19 by virologists, policymakers and IA researchers. | The paper has taken initial steps in compiling and highlighting existing state-of-the-art, but does not discriminate between working cases in wild and in laboratory circumstances. |
| [44] | COVIDX-Net, VGG19 and (DenseNet) | f1-scores of VGG19 is 0.89% and DenseNet is 0.91% | The technique allows radiologists to detect COVID-19 instantly in X-ray images | X-ray scans cannot differentiate between the soft tissue and the medium dose to minimize exposure to the patients |
| [43] | Convolutional Neural Network (CNN) and CLSTM-based deep learning models | 91% accuracy for the ConvLSTM DLMs and the CNN and | The method Improved the learning capacities of the Convolutional Neural Network (CNN) and CLSTM-based deep learning models (DADLMs) by introduced a two machine learning models to in order to enhance the prediction accuracy of COVID-19 identification. | The method has been investigated under two machine learning techniques SVM and k-NN only. |
| [45] | Deep learning model | 92.4% classification accuracy | The model classified and define groups as regular, non-COVID, and COVID-19 based on a deep learning model with accuracy of 92.4% | The method is need to be improved in term of defining and classification accuracy. |
| [47] | CNN | 98.3% accuracy and a precision of 96.72% | This model identifies Coronavirus patients with very little time and energy, and is CNN models very accurate. In their work, the CNN models in COVID-19 are also studied in a comparative analysis. | The study preserving the dataset images with transformed to $224 \times 224$ pixels due to the image quality. Actually, the converting process is considered as an extra step before using CNN model. |
| [48] | ResNet50 plus SVM | The accuracy of SVM scored, 95.38%, 91.41%, 95.52%, 90.76% for FPR, MCC, F1-score, and Kappa respectively for COVID-19 detection | The method identified the features extracted from various CNN models using X-ray images and employed a support vector machine (SVM). Their analysis notes the highest results of the ResNet50 model with the SVM classifier. | The method was used the SVM only rather than other machine learning techniques. |
| [52] | FrMEMs approach | The classification accuracy scores with 96.09% and 98.09% for the COVID-19 datasets. | The model used the FrMEMs approach to take advantage of chest X-ray images features. The computing process was accelerated using a parallel multi-core computational architecture. | The limitation of the method is that the time of the CPU is considered as the third rank. |

managed by LWR approaches, and boundary and cluster impacts can be avoided. LWR depends on how far the nearest neighbours of a given query example are retrieved from a function. Nevertheless, the distance function does not have to follow the formal distance metric requirements [52]. The RL allows several ways to use distances [52]; for example, a function for a single instance is used in all parts of the input space (global distance function); (ii) parameters of the distance function are determined by a process of optimization (request-based local distance function), or (iii) a distance function and its parameter values (point-based local distance function) are provided for every training example. Weighting and smoothing parameters are also relevant for LWR. A weighting function (kernel) determines the weight of a query example by a neighbour. The maximum value of a weighting function should be zero and decay smoothly with increasing distance. Examples of well-known weighing functions are Cubic, Epanchnikov, Tricube, Inverse and Gaussian. In terms of smoothing parameters, the

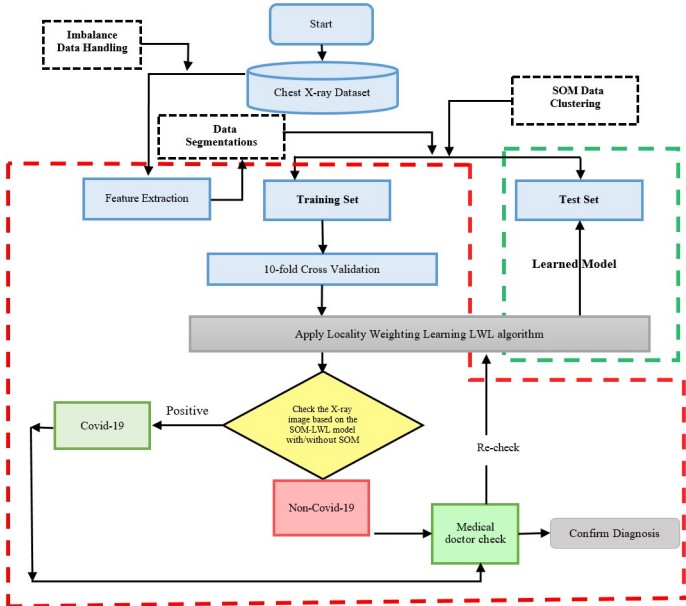

**Fig 1. Operational framework.**

parameter (h) of the bandwidth determines the size or spectrum of the generalization. There are many ways to describe parameter *h* [52], for example, by selecting a fixed bandwidth, choosing the next neighbour bandwidth, choosing a regional bandwidth, selecting a local query-based bandwidth or selecting a local point-based bandwidth. In favour of the closest bandwidth selection approach, Cleveland & Loader [56] argued to determine the value of h; in this case, parameter h was equal to the distance from a *k-th* example.

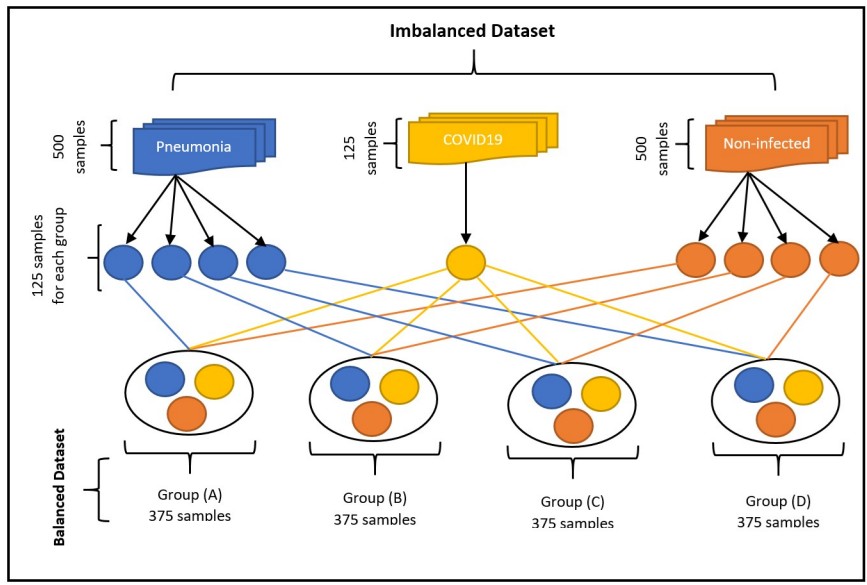

**Fig 2. Imbalance data handling process.**

### 3.3. Evaluation measures

This section discusses the evaluation measures that are used to assess the performance of the proposed method, which are as follows:

**3.3.1. Mean absolute error (MAE).** Sum used to assess how close the predictions or forecasts are to the actual outcomes. Examples for *Y* versus *X* provide measurements of the prediction versus the actual time versus initial time as well as a measurement technique versus an alternative measurement technique. The MAE is calculated as:

$$MAE = \frac{1}{N} \sqrt{\sum_{t=1}^{n} |t_e|} \tag{1}$$

**3.3.2. Root mean square error (RMSD).** Measures the variations between the values expected by a model or estimator (sample and population values) and the values actually observed. It represents the standard sample deviation between the values expected and the values observed. It adds to one predictive power of the size of the errors in predictions for various periods. It is a fair measure of precision but is only used to evaluate forecast errors in different models for a certain variable and not between variables, since it depends on the scale. It is also named the root mean square deviation (RMSD), the RMSE is calculated as:

$$RMSE = \sqrt{\frac{1}{n} \sum_{t=1}^{n} e_t^{\,2}} \tag{2}$$

**3.3.3. Relative absolute error (RAE).** The relative squared error refers to a situation where a simple predictor is used. This simple predictor is more specifically just the average of actual values. Thus, the relative squared error takes and normalizes the entire squared error and divides it by the simple predictor's total squared error. Relative squared error (Ei) is assessed mathematically as:

$$E_i = \frac{\sum_{j-1}^{x} |P_{ij} - T_j|}{\sum_{j-1}^{x} |T_i - \bar{T}|} \tag{3}$$

*where P(ij) is the parameter for sample case j for the particular program I (out of n sample cases). The sample case J class parameter is Tj:*

$$\breve{T} = \frac{1}{n} \sum_{j-1}^{x} T_j \tag{4}$$

The number must be equivalent to 0 and *Ei* = 0 to fit perfectly. The egg index therefore varies between 0 and infinity, with 0 matching the ideal. The relative absolute error is somewhat similar to RSE in the sense that it is also related to a simple indicator that is just the average of the actual value. In this case, however, the error is the absolute total error, rather than the complete squared error. Therefore, the absolute relative error takes the total absolute error and normalizes it by separating the actual total error from the basic predictor.

**3.3.4. Root relative squared error (RRSE).** The RSF refers to what the error would have been if there had been a simple predictor. More precisely, this basic measure is just the average

real value. Thus, the relative squared error takes and normalizes the squared error, dividing it by the simple predictor's total squared error. When taking the square root of the comparable squared error, the error is of the same dimensions as the expected number. Statistically, the RRSE Ei of a distinct model i is assessed by the following calculation:

$$E_i = \frac{\sum_{j-1}^{x} (P_{ij} - T_j)^2}{\sum_{j-1}^{x} (T_i - \bar{T})^2} \tag{5}$$

*where P(ij) is the expected parameter for sample case J by the separate program I (from n samples); Tj is the class parameter for sample case j; and $\bar{T}$ is specified as defined in* Eq 4.

**3.3.5. Correlation coefficient (CC).** The correlation coefficient is a statistical measure of the relation intensity between two variables' relative movements. The values differ from -1.0 to 1.0. An error in the correlation calculation is a measured number greater than 1.0 or less than -1.0. A correlation of -1.0 is completely negative, while a correlation of 1.0 is completely positive. A correlation between 0.0 and the movement of the two variables does not appear to be linear (see below):

$$r = \frac{n(\sum xy) - (\sum x)(\sum y)}{\sqrt{[n\sum x^2 - (\sum x)^2][n\sum y^2 - (\sum y)^2]}} \tag{6}$$

*Where n is sample size, x and y are the specific sample points indexed with i.*

## 4. Methodology and approach

### 4.1. Feature extraction procedure

By examining X-rays, we can see that good texture and statistical groups are possibly the principal visual attribute. Several researchers have started using texture and statistical features over the last decade to identify models for classification problems. This type of function has become a major trend because it can be easily done, as the software engineering work is usually a laborious job and involves a sophisticated knowledge of problem classes, and the techniques supporting hand design descriptors are not essential. This function is not essential. Although the non-manufactured descriptors have some obvious features, we should note that the handmade characteristics have specific characteristics that can also make them very useful for coping with many classification tasks. One of these benefits is that handmade features are more robust since these techniques are often operating in a more deterministic manner to capture trends relating to the problem. Rather than using uncrafted features, a more accurate interpretation of patterns produced by handcrafted features of the pictures is more feasible. Nevertheless, in this work, we have made efforts to use these two groups in extracting features. In this way, we can test the two separately, and we conduct a combination of several experimental set-ups. In this sense, we make use of the complementarity between the two descriptors' strategies, since they do not necessarily make the similar mistakes in the performance of a specified prediction task, as demonstrated in [57, 58]. In this section, the descriptors utilized for this study are briefly listed. The selected texture descriptors were chosen to achieve good results in common applications or precisely in medicinal image investigation systems. The statistical features group includes the following:

**4.1.1. Mean.** The mean is a measure of the average intensity of the neighbouring pixels of an image.

$$\mathbf{m} = \sum_{i=0}^{l-1} z_i * p(z_i) \tag{7}$$

**4.1.2. Standard deviation.** The standard deviation is a measure of how spread out numbers are.

**4.1.3. Skewness.** The skewness, or more specifically, lack of symmetry, is a measure of symmetry. If the left and right points around of the middle are identical, then the distribution or dataset is symmetric. The skewness is zero for a regular distribution, and any symmetric data should be near zero. Negative skewness values indicate left skewed data, and right skewed data indicates positive skewness.

$$\textbf{\textit{Skewness}} = \sum_{i=0}^{l-1} (z_i - m)^3 * p(z_i) \tag{8}$$

**4.1.4. Kurtosis.** The kurtosis is a measure of whether the data in relation to normal distribution are peaked or flat. In other words, high-kurtosis datasets appear to have a distinct peak close to the average.

$$\textbf{\textit{Kurtosis}} = \sum_{i=0}^{l-1} (z_i - m)^4 * p(z_i) \tag{9}$$

**4.1.5. Contrast.** The contrast is the luminance and/or colour difference that distinguishes the item (or its display in the picture or display). In real-world visual perception, the difference in the colour and luminosity of the target and other objects in the same field of view defines the contrast.

$$\textbf{Contrast} = \sum_{i=0}^{l-1} \sqrt{(z_i - m)^2 * p(z_i)} \tag{10}$$

**4.1.6. Smoothness.** Smoothness measures the relative intensity variations in a region.

$$\textbf{Smoothness} = 1 - \frac{1}{(1 + \sigma 2)} \tag{11}$$

*where zi is a random intensity indicator variable, p(z) is the histogram for the levels of intensity of the field, l is number of potential intensity levels and σ is standard deviation factor.*

For the texture features group, we applied the GLCM method. Introduced by Haralick [59], GLCM is a connection between pixels in a matrix that is frequently used in the analysis of texture. Adjacency is a relation between two pixels that is defined by the distance between the two pixels and the angle between them. The size and angles of the space are therefore GLCM parameters. The GLCM functions describe the textures of an image by measuring how many pixel pairs occur in an image with certain values and with a given spatial relation, and a GLCM is generated. Then, statistical measures are obtained from the matrix pair of pixels with different values and in a given spatial relation. We noted that in the texture features, the statistical measures of texture filter functions cannot give information on the shape, i.e., the spatial relationships of pixels in the image. The GLCM feature set is based on second-order statistics. The overall average for degrees of similarity between pixel pairs in different ways (homogeneity, uniformity, etc.) can be used for the reflection. One of the key factors influencing GLCM's

**Table 2. GLCM texture features formulas.**

| Sl. No. | GLCM feature | Formula |
|---|---|---|
| 1 | Correlation | $\sum_{i,j=0}^{N-1} P_{i,j} \left[ \dfrac{(i - \mu_j)}{\sqrt{(\sigma_i^2)(\sigma_j^2)}} \right]$ |
| 2 | Homogeneity | $\sum_{i,j=0}^{N-1} \dfrac{P_{i,j}}{1 + (i+j)^2}$ |
| 3 | Energy | $\sum_{i,j=0}^{N-1} P_{i,j}(-\ln P_{i,j})$ |
| 4 | Contrast | $\sum_{i,j=0}^{N-1} P_{i,j}(i - j)^2$ |

capacities for discrimination is the pixel separation. When taking the distance as 1, the association between pixel values (i.e., short-term neighbourhood connectivity) is expressed. The change in the value of the distance represents how much pixels correspond.

**4.1.7. GLCM features.** In 1979, Haralick suggested 14 characteristics in "Statistical and structural texture approaches" [59], indicating that functions that well describe the adjacency relations among pixels in the image texture are produced by the GLCM. The characteristics extracted by some formulas from co-occurrence matrices depend on features to be observed. We selected four of the Haralick texture features based on the X-ray image dataset characteristics, such as correlation, homogeneity, energy and contrast. Table 2 shows some formulas to compute GLCM texture features.

## 4.2. Self-organization map

In the 1980s, the SOM was launched by Teuvo Kohonen from Germany; it is often known as a Kohonen map. The algorithm is a kind of artificial neural network that is learned to generate a small (typically two-dimensional) non-supervised learning representation of the sample input field, called a map, and thus is a tool for that dimensionality. SOMs are distinct from other artificial neural networks because competitive training is used in contrast to error correction (e.g., gradient descent and back propagation) and they maintain the place's topological qualities by using a neighbourhood function. One essential detail is that the entire training takes place without control, i.e., the nodes are structured themselves. They are often named feature maps, and the characteristics of the input data are basically retrained and clearly grouped according to similarity. The map has a logical value for the visualization, in a small, usually two-dimensional area, of huge quantities or complex of high-dimensional data to determine how it is defined by the given unlabelled data. The Kohonen map [60] is an unmonitored learning calculation to generate the topology-conserved changes from a high-dimensional data space to a small-guided space and is a capable apparatus that is used in a variety of fields, such as knowledge mining, analysis, perception and grouping. SOM uses have grown into different fields, such as online research [61], bioinformatics [62] and back-propagation neural network methods [63], and their value continues to increase. Because of the increasing importance of the SOM and its development, only vector-based knowledge can be handled. In the event of a dataset without a vector, the information must be vectorized or adjusted to the data composition of the Kohonen itself. In that way, the Kohonen family presents an unavoidable question in terms of determining the autonomous representation of the written knowledge in the Kohonen calculation. The Kohonen Map Architecture [64] is demonstrated in Fig 3.

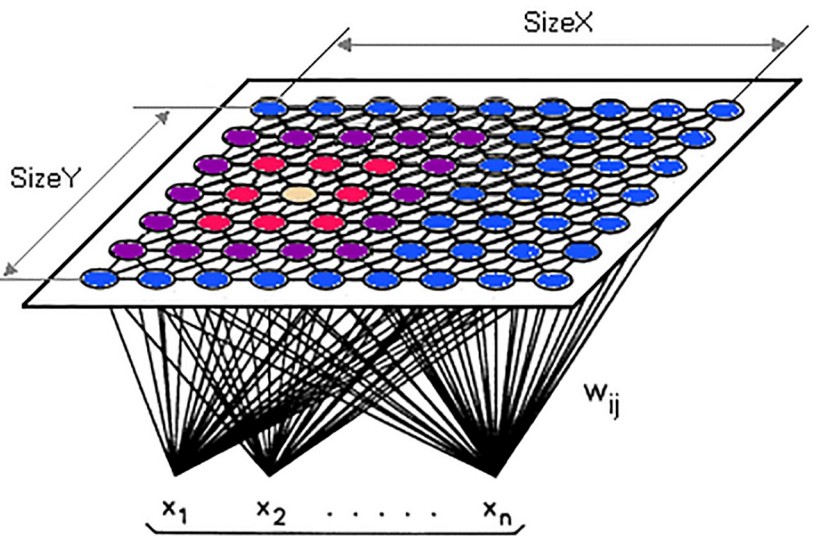

**Fig 3. Kohonen map architecture [64].**

Fig 3 shown the main structure of the Kohonen Map clustering technique that has been used in the proposed method.

"*w is the weight vector*

*$w\_ij(t)$ is the weight of the connection between the nodes i,j in the grid, and the input vector's instance at the iteration t*

*x is the input vector*"

**Kohonen Map Algorithm**

```
Begin
1: Set a random value for each node's weight w_ij
2: Use a random vector data x k
3: Repeat point 4. and 5. For all map nodes:
4: Calculate the Euclidean distance to wij, the weight vector of first
node, and the input vector x(t), where t, I j = 0.
5: Follow the node with the distance that yields the smallest t.
6: Select the overall best-matching unit (BMU), i.e., the node with
the smallest distance from all determined.
7: Determine the BMU radius of βij(t) topological neighbourhood in the
Kohonen Map
8: Replay the vector weight w j of the first node in the BMU district
by adding a fraction of the difference between nodes in the BMU
district
9: Step 1 is the initialization stage, whereas steps 2 to 9 define the
learning stage.
End
```

Updates and changes to the variables are made as follows:

$$w_{ij}(t+1) = w_{ij}(t) + \alpha_i(t)[x(t) - w_{ij}(t)] \tag{12}$$

Or

$$w_{ij}(t+1) = w_{ij}(t) + \alpha_i(t)\beta_{ij}(t)[x(t) - w_{ij}(t)] \tag{13}$$

*The first formula informs us that the new wij (t + 1) for node I j is the same as the sum of old w ij(t) and the difference is a small fraction of the old wij(t) weight. The weight vector is "moved"*

*to the input vector in other words. Another essential factor is that the updating weight of the nodes in the neighborhood radius should be proportionate to the 2D size.*

## 5. Experimental design and datasets

This section describes the experimental setup and assessment of the suggested LWL based on the SOM diagnostic method. Computation performance addresses the influence of the suggested process on its performance and calculation steps. The experiments are all carried out in the MATLAB 9.3 Release R2017b environment, IBM SPSS modular and Weka 8.3 tools.

### 5.1. Experimental design

In our suggested method, we enhance the existing X-ray dataset using crossover balanced COVID-19 class images. Our aim here is to demonstrate the negative impact of the imbalanced distributions in the raw dataset on performance. It is worth mentioning that we adjust the SOM-LWL for a regular training process with the best model parameters. The research aims at introducing a prediction method for COVID-19 diagnoses using a hybrid SOM clustering algorithm and LWL method for improving the diagnostic precision of the classification and reducing the misdiagnosis error. This research leads to a new approach that blends supervised and unsupervised methods of learning as a hybrid model. A qualified study was carried out using the LWL classification and SOM clustering data structure on the X-ray chest image feature extraction. The outcomes of the clusters were used as inputs to the classification model by using LWL as predictions for positive cases of COVID-19, pneumonia, and no-findings cases of COVID-19. The methodology of the hybrid SOM-LWL was used to test the effects of the qualified process. The data were associated with multiple cases (non-COVID-19, pneumonia, and COVID-19). For the training and testing of the SOM-LWL method, the dataset was divided into 10 portions based on 10-fold cross validation.

Cross validation involves the simple idea of holdout by using certain information for testing and the rest for training. Repeated holdout enables the use of more data in training than in testing while still providing a reliable test. In 10-fold cross validation, the contents of one fold are influenced by the contents of other folds. The different cross-validations are applied with independent samples from the COVID-19 dataset to obtain some variations in results and remove any outliers based on averaging. The COVID-19 dataset was divided into 10 folds after the data balancing process. Two different scenarios have been used for the identification and classification of COVID-19 in X-rays. First, the SOM-LWL scheme is trained to classify the X-rays into three classes: COVID-19, Non-COVID-19, and pneumonia. Furthermore, two classes are trained with the SOM-LWL model: the COVID-19 classes and the Non-COVID-19 classes. For triple and binary classification problems, the output of the suggested model is assessed by the 10-fold cross validation process. The training records use 90% of the X-ray images and 10% as the testing stage, the process is performed four times based on the balance of the dataset that has been determined in pre-processing phase. In contrast to the traditional hold-out validation process, this type of validation method provides better results.

### 5.2. The dataset

AI-based X-ray screening is effective in both asymptomatic and symptomatic patients for COVID-19 testing. A unique challenge for algorithms is that COVID-19 can be distinguished from other lower respiratory diseases that may look similar in X-ray imaging. The data are produced in.png, jpg, and jpeg X-ray formats. A collection by Dr. Cohen of John Hopkins Hospital uses the two datasets from the Kaggle Chest X-rays [65]. These datasets were used to compare cases with bacterial pneumonia, healthy cases and cases with pneumonia induced by

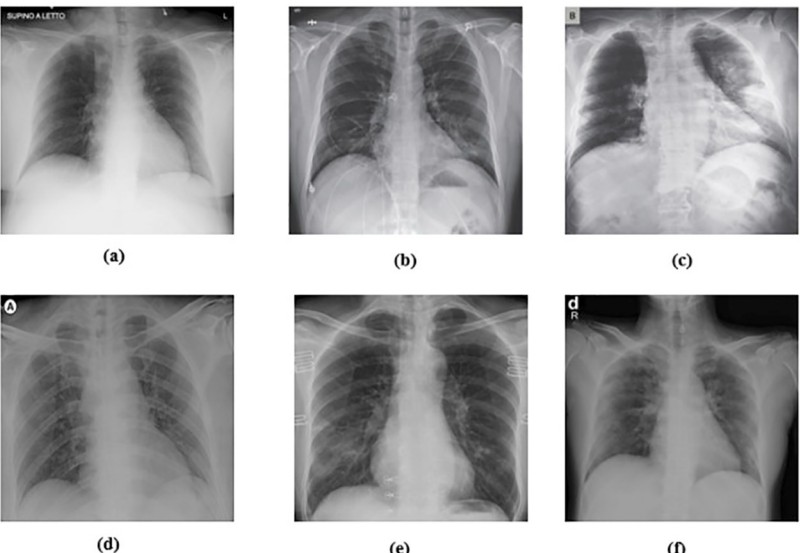

**Fig 4. X-ray sample from the COVID-19 dataset [23].**

COVID-19 viruses. The dataset is made up of chest images of pneumonia patients. Cohen JP [65] developed a COVID-19 X-ray image database using images from different open access sources. This database is continuously updated with images from various regions shared by scientists. The database currently has 127 COVID-19 diagnosed X-ray images. Fig 4 demonstrates some COVID-19 samples acquired from the X-ray dataset.

Fig 4 shown some sample of COVID-19 cases that has been acquired from the X-ray dataset.

Cohen compiled and collected COVID-19 images from various public outlets. From this database, a total of 88 positive cases were obtained. Fig 4 (top) displays the COVID-19 sample images from this database. However, the database does not include usual (negative) events. Fortunately, regular chest X-ray images are widely available.

Within the sample, there are 43 females and 82 males who have been shown to be positive. Complete metadata are not provided for all patients in this dataset. The age of 26 positive COVID-19 individuals is given, and their average age is approximately 55 years. In addition, for normal and pneumonia images, a database of ChestX-ray8 has been provided by Wang et al. [66]. To prevent unbalanced results, we used the random images of this set of 500 no-findings and 500 pneumonia frontal chest X-rays. The classes of groups produced were equivalent such that each group consisted of 375 cases consisting of 125 COVID-19 cases labelled as class 1, and 125 non-COVID19 infected cases and 125 pneumonia cases, respectively, were labelled as class 2 and class 3. Through this method, the generated groups after data balancing were individually identified in diagnostic cluster experiments for each group. To compare the correlation factor of the X-ray diagnosis classifier, experimentations were conducted using the SOM-LWL learning classifier with 10-fold cross-validation. The new balanced datasets were divided into 10 pieces. Each part accounted for 10% of the original dataset, such that each dataset set could be used as test data. In every round, nine sets of experiments are used for training and one for testing. The SOM approach is used to cluster the chest X-ray dataset based on non-COVID-19, pneumonia, and COVID-19 characteristics of the same type.

## 6. Results discussion and analysis

The error is linked to what a classifier would have been. A simple classifier provides the average true values obtained from the learning data. Therefore, relative squared error assumes and normalizes the overall squared error by dividing the default predictors by a minimum squared error. To evaluate our X-ray COVID-19 identification model, the mean absolute error, root mean squared error, relative absolute error, correlation coefficient, and root relative squared error have been calculated as standard measures, which has been discussed in Section 3.

The performance of the suggested SOM-LWL model is tested with the chest X-ray COVID-19 dataset. The results of the SOM algorithm are the extracted 12 clusters with various instances and characteristics distributed based on the image feature extraction. With the criterion defined for the grouping criterion, the SOM algorithm determines the best number of clusters automatically. Fig 5 shows the generated clusters.

Fig 5 demonstrates the generated clusters using SOM algorithm with various instances and characteristics distributed based on the image feature extraction.

Furthermore, in the latter part of the training, when the SOM-LWL model continuously analyses all X-rays at each point during the training, these quick oscillations become sluggish. The performance of multi-class prediction and average classification of the SOM-LWL model has been calculated and estimated for all folds. In Fig 5, SOM clustering algorithm outcomes generated 12 clusters. The range of the distributed percentage of cases members is between 0.1% and 16.13%. Due to the similarity of the features, the similar cases are reported in Cluster 1, Cluster 2, etc. The number of the members that is the highest is scored with 223 instances and represents 19.8% of the total instances. The ratio of the largest size to the smallest cluster is scored with 223 instances, as shown in Fig 5. The similarity and diversity of these clusters is highlighted in the dataset instances, consequently helping to identify variations among members of the dataset and facilitating the classification and learning process when constructing the LWL diagnostic model. Using these clusters, a chest X-ray dataset was analysed and represented by the SOM clustering algorithm; data interpretation is the principal task of many of the clustering processes. It is one of the reasons for choosing the hybrid approach of the SOM clustering algorithm and LWL classification, though data prediction is the main task of the classification technology. Regarding the stage of how the clustering is used and combined with the LWL classification for grouping of datasets in various groups, the SOM clustering algorithm is first applied. Within a new variable function called Label, the outputs of these classes and clusters are represented. We plan to validate our model in the future with the inclusion of additional images. This built model can be used as a cloud implementation, so that patients can be immediately identified and rehabilitated using the SOM-LWL model. This could significantly reduce the workload of the clinician.

The chest X-ray image data collection was studied for the purpose of performing an experimental analysis. As previously reported, the study employed a 10-fold method of cross-

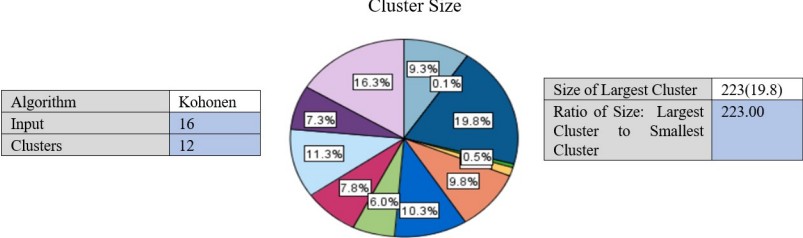

**Fig 5. X-ray sample of COVID-19 dataset.**

**Table 3. Results of chest X-ray COVID-19 classification based on LWL-SOM.**

| Experiment No | Samples cases | Correlation coefficient | Mean absolute error | Root mean squared error | Relative absolute error | Root relative squared error | Total Number of Instances |
|---|---|---|---|---|---|---|---|
| | | Locally weighted learning (LWL) with data clustering using self-organized mapping (SOM) | | | | | |
| **Experiment -1** | Covid19 vs No finding | 1 | 0 | 0 | 0% | 0% | 625 |
| **Experiment -2** | Covid19 vs Pneumonia | 0.9999 | 0.0018 | 0.0067 | 0.55% | 1.6716% | 625 |
| **Experiment -3** | Pneumonia vs No finding | 1 | 0 | 0 | 0% | 0% | 1000 |
| **Experiment -4** | Covid19 vs No finding vs Pneumonia | 0.9788 | 0.1009 | 0.2 | 11.3365% | 21.1972% | 1125 |

validation for training and dataset testing of the balanced datasets. To study the resulting improvement with the hybrid approach, the experiment was carried out using an LWL classification with and without clustering results.

To measure the effect of COVID-19, a chest X-ray information dataset was extracted and analysed. The dataset was identified positive COVID-19, Non-COVID-19, and pneumonia cases for each patient. The hybrid technique employed the mixture of SOM and LWL methods for learning and testing the dataset. The dataset was then divided into multiple clusters of various instances using the SOM algorithm. The key goals of the research were to derive patterns and structures by collecting samples of the same characteristics and features of COVID-19, thus decreasing the difficulty of accurate diagnostics. Tables 3 and 4 provide a set of results obtained with the LWL classification method without and with clustering using the SOM approach for the training and testing experiments. The performance of SOM is evaluated in the combination phase as a new function that identifies each instance on the cluster name dataset, as demonstrated in Section 5. By grouping the dataset into similar clusters, this feature may aid the association between instances. The LWL classifier was again used to achieve a high correlation factor with the high performance of the SOM process. In training and testing with and without clustering, 10-fold cross validation was applied to examine the integrated features of the clustering process with extracted features from the chest X-ray image dataset. Each testing and training experiment chose the images features extracted as an input variable to the LWL classifier. The class field is the target (COVID-19, Non-COVID-19, and pneumonia) cases. When the LWL technique classified the instances of the dataset with the SOM cluster output, the correlation results were increased, and the classification error decreased accordingly. Importantly, the SOM clustering method increased the correlation factor by a ratio (0.978) for all cases COVID-19, Non-COVID-19, and pneumonia cases, a (1) ratio correlation between the pneumonia and Non-COVID-19 cases, a (0.990) ratio for the COVID-19 and pneumonia cases, and a (1) ratio between the COVID-19 and Non-COVID-19 cases, as shown in Tables 3 and 4.

**Table 4. Results of chest X-ray COVID-19 classification based on LWL.**

| Experiment No | Samples cases | Correlation coefficient | Mean absolute error | Root mean squared error | Relative absolute error | Root relative squared error | Total Number of Instances |
|---|---|---|---|---|---|---|---|
| | | Locally weighted learning (LWL) without data clustering | | | | | |
| **Experiment -1** | Covid19 vs No finding | 0.8894 | 0.0597 | 0.1831 | 18.6029% | 45.6483% | 625 |
| **Experiment -2** | Covid19 vs Pneumonia | 0.8783 | 0.0694 | 0.1913 | 21.6775% | 47.7803% | 625 |
| **Experiment -3** | Pneumonia vs No finding | 0.6113 | 0.3121 | 0.3957 | 62.3354% | 79.0202% | 1000 |
| **Experiment -4** | Covid19 vs No finding vs Pneumonia | 0.9613 | 0.1352 | 0.2621 | 15.1846% | 27.7699% | 1125 |

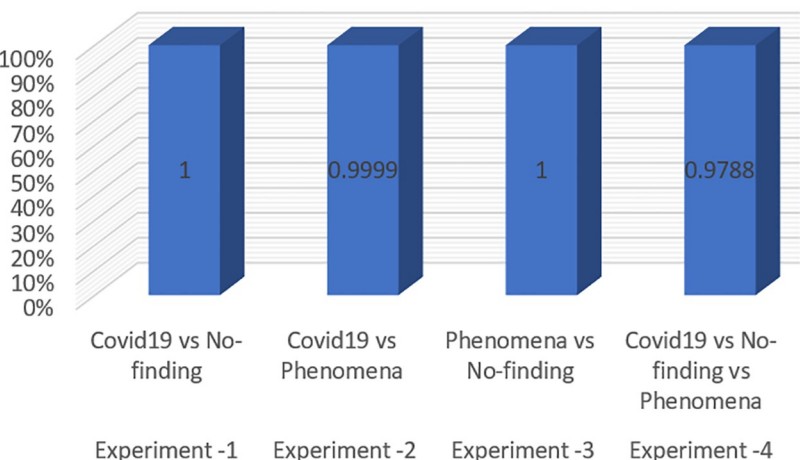

**Fig 6. SOM-LWL correlation coefficient.**

The method used in the SOM-LWL model produced an increase in the correlation coefficient results between the Covid19, no-finding, and pneumonia cases; pneumonia and no-finding cases; Covid19 and pneumonia cases; and Covid19 and no-finding cases from 0.9613 to 0.9788, 0.6113 to 1 0.8783 to 0.9999, and 0.8894 to 1, respectively. Moreover, using the suggested model, decreases in the mean absolute error, root mean squared error, relative absolute error, and root relative squared error were progressively achieved for the three best results with low error ratios when using SOM clustering algorithm with the LWL classifier. We noted that the experiments conducted used different dataset sizes according to data balancing among the types of chest X-ray image cases. The individual results for the SOM-LWL correlation coefficient are demonstrated in Fig 6.

Figs 6, 7, 8, 9 and 10 demonstrate the output results of the LWL with clustering using the SOM method using different evaluation criteria such as mean absolute error, root mean squared error, relative absolute error, and root relative squared error.

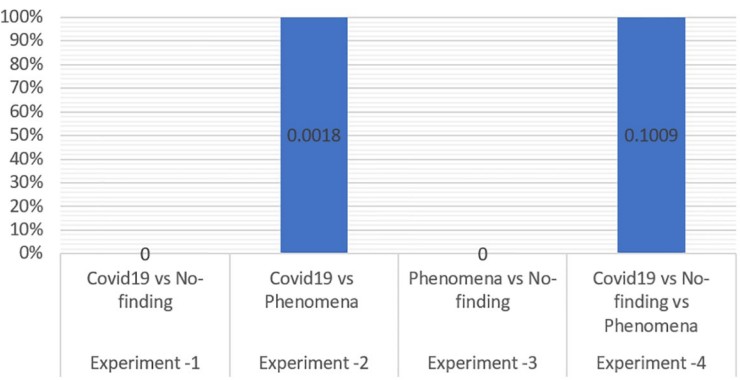

**Fig 7. SOM-LWL mean absolute error.**

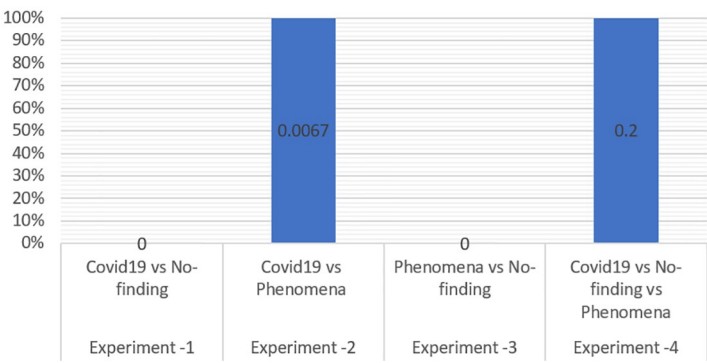

**Fig 8. SOM-LWL root mean squared error.**

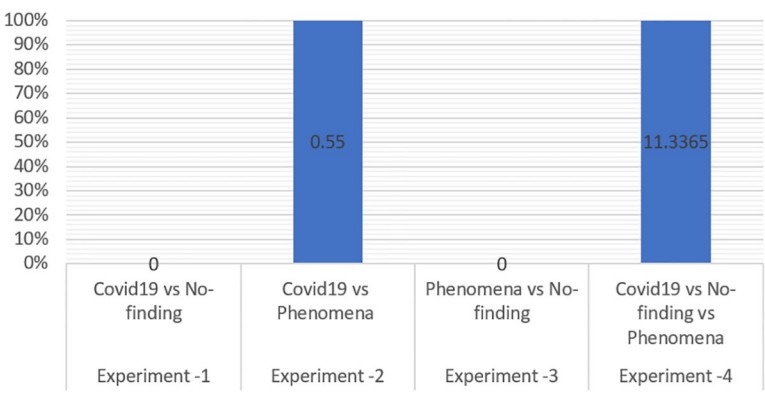

**Fig 9. SOM-LWL root relative error.**

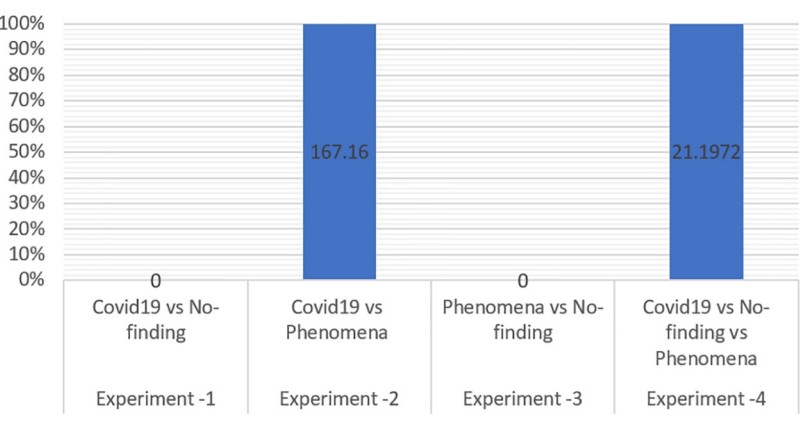

**Fig 10. SOM-LWL root relative squared error.**

**Table 5. T-test statistical significance results.**

| | Differences between the correlation coefficient factor, mean absolute error, root mean squared error, relative absolute error, and root relative squared error before and after the improvement | | | | t | df | P Value |
|---|---|---|---|---|---|---|---|
| | Mean | Std. Deviation | 95% Confidence Interval of the Difference | | | | |
| | | | Lower | Upper | | | |
| Correlation coefficient factor | -.15960 | .14787 | -.28322 | -.03598 | -3.053 | 7 | .019 |
| Mean absolute error | .11843 | .12026 | .01788 | .21897 | 2.785 | 7 | .027 |
| Root mean squared error | .20638 | .12837 | .09906 | .31369 | 4.547 | 7 | .003 |

The correlation coefficient factors were calculated, and the correlation coefficient scores using LWL with clustering achieved high factors with error scores of 1 and 0 for COVID-19 with non-COVID-19 and pneumonia vs non-COVID-19. The Figures also indicate that better results are achieved with the LWL classifier with SOM clustering than with the LWL without the clustering approach yielding (0.9999) and 0.9788 correlation coefficient factors between the COVID-19 vs pneumonia cases, and COVID-19 vs Non-COVID-19vs pneumonia cases, respectively. High-performance results without clustering are achieved in COVID-19, Non-COVID-19, and pneumonia sample cases with a (0.9613) correlation coefficient factor. On the other hand, high-performance diagnosis results with clustering are obtained in the same sample cases (COVID-19, Non-COVID-19, and pneumonia) with a score of (1) for the correlation coefficient factor. We concluded that there is an improvement while using the SOM clustering method. The prediction results of the SOM-LWL with clustering are better, and the COVID-19 diagnosis is more precise when using an integration of the SOM output with the LWL classifier.

The results of our prediction model experiments showed enhancements were obtained by the SOM-SVM model, and the t-test algorithm was used as the statistical significance measure to emphasize the improvement. The low t-test values (typically less than 0.05) indicate that the two variables are substantially modified. This condition was highlighted in the assessment measures based on the findings achieved in Table 5 concerning the correlation coefficient factors, mean absolute error, and root mean squared error values of 0.019, .027, and.003, respectively. This reveals that SOM-LWL achieved significant enhancement in diagnostic performance, and the LWL with and without clustering is substantially different. Table 5 demonstrates the performance results using the t-test statistical significance test.

We noted that in Table 5 the P-value score is less than 0.05, thus indicating that for the two variables, the correlation coefficient factor, mean absolute error, and root mean squared error improved significantly after using the SOM clustering method.

The comparison of the suggested SOM-LWL scheme with other COVID-19 diagnostic systems developed based on chest X-ray is demonstrated in Fig 11.

Fig 11 shows a summary of the comparison performances between the proposed SOM-LWL method and other COVID-19 chest X-ray diagnosis methods.

We note the achieved better results in the multi-class prediction scenario (COVID-19, Non-COVID-19, and pneumonia). In addition, the performance results have proven that in terms of the diagnostic monitoring for the early diagnosis, treatment and incubation phases of the disease, radiological imaging plays an important role in the COVID-19 epidemic.

## 7. Conclusion and future work

A few characteristic findings in the lungs of patients with COVID-19 can be identified by chest X-rays. In this study, the SOM-LWL model is suggested for diagnosis and detection of the

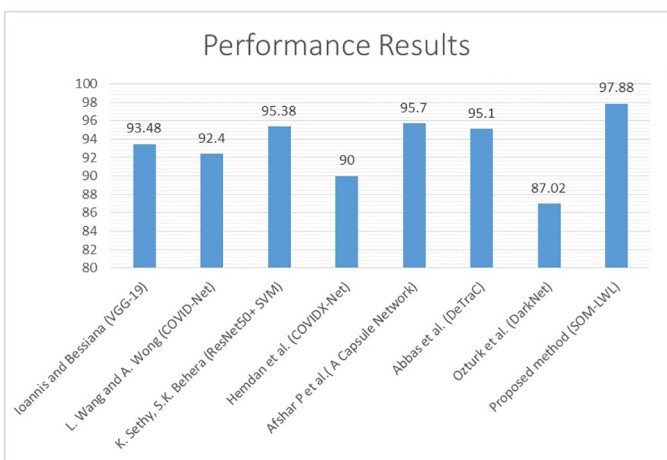

**Fig 11. Comparison of the proposed SOM-LWL method with other COVID-19 diagnostic methods.**

COVID-19 disease based on chest X-rays. The number of cases continues to rise exponentially as COVID-19 spreads across the world. To prevent crippling the healthcare system, the use of a tool that can help diagnose the disease in people by using an inexpensive and fast process is necessary. Within this context, the literature suggests that the diagnosis may be assisted by the use of data mining methods to classify pneumonia disease in chest X-rays. However, the issue is much more difficult when we look at chest images of patients suffering from pneumonia caused by multiple types of pathogens and attempt to forecast a particular form of pneumonia (COVID-19). There are far more people without pneumonia than people who are sick in the real world. Moreover, the number of people suffering from pneumonia caused by various pathogens is inherently imbalanced, and due to the COVID-19 outbreak, it is increasingly difficult to measure the precise imbalance between these numbers. In view of a plausible scenario, we have suggested a classification scheme to classify and define COVID-19 as a pneumonia disease caused by various pathogens in chest X-rays. We use resampling methods in the proposed method to counter the problem's inherent imbalance. In addition, the conceptual scheme includes 8 separate sets of features derived from the images that are evaluated separately and subsequently integrated in an early fusion design. In addition, exclusively and in a late fusion configuration, the prediction outputs are tested. The suggested schema also implements multi-class, unsupervised learning (SOM clustering) and supervised learning (LWL). To apply the diagnosis model in this application field, we have considered a prediction model called SOM-LWL.

In the future work, the proposed method will be expanded to be abdicable for different types of COVID-19 datasets such as SARS-CoV-2 CT-scan [67], COVID-CT [68], and statistical datasets. However, the quality of predication method in COVID-19 disease will be combined with optimization techniques using classification and regression algorithms.

## Author Contributions

**Conceptualization:** Ahmed Hamza Osman, Hani Moetque Aljahdali, Sultan Menwer Altarrazi, Ali Ahmed.

**Data curation:** Ahmed Hamza Osman, Ali Ahmed.

**Formal analysis:** Ahmed Hamza Osman.

**Funding acquisition:** Ahmed Hamza Osman.

**Investigation:** Ahmed Hamza Osman.

**Methodology:** Ahmed Hamza Osman, Hani Moetque Aljahdali, Ali Ahmed.

**Project administration:** Ahmed Hamza Osman, Hani Moetque Aljahdali.

**Resources:** Ahmed Hamza Osman, Ali Ahmed.

**Software:** Ahmed Hamza Osman.

**Supervision:** Ahmed Hamza Osman.

**Validation:** Ahmed Hamza Osman, Hani Moetque Aljahdali, Sultan Menwer Altarrazi.

**Visualization:** Ahmed Hamza Osman.

**Writing – original draft:** Ahmed Hamza Osman.

**Writing – review & editing:** Ahmed Hamza Osman.

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
