## [Decision Letter · Decision Letter 0]

28 Oct 2020

PONE-D-20-32000

SOM-LWL Method for Identification of COVID-19 on Chest X-rays

PLOS ONE

Dear Dr. Osman,

Thank you for submitting your manuscript to PLOS ONE. After careful consideration, we feel that it has merit but does not fully meet PLOS ONE’s publication criteria as it currently stands. Therefore, we invite you to submit a revised version of the manuscript that addresses the points raised during the review process.

In particular:

characteristics of included patients are not clearly provided,analysis of recent related papers is missing,figures 1, 5 and 11 are not clear,it is not clear if cross-validation scheme was used,proofreading is required to fix typos.

We look forward to receiving your revised manuscript.

Kind regards,

Maciej Huk, Ph.D.

Academic Editor

PLOS ONE

Journal Requirements:

4.  Please ensure that you refer to Figure 3 and Figure 10 in your text as, if accepted, production will need this reference to link the reader to the figure.

Reviewers' comments:

Reviewer's Responses to Questions

**Comments to the Author**

1. Is the manuscript technically sound, and do the data support the conclusions?

Reviewer #1: Partly

Reviewer #2: Partly

2. Has the statistical analysis been performed appropriately and rigorously? 

Reviewer #1: Yes

Reviewer #2: Yes

3. Have the authors made all data underlying the findings in their manuscript fully available?

Reviewer #1: No

Reviewer #2: Yes

4. Is the manuscript presented in an intelligible fashion and written in standard English?

Reviewer #1: Yes

Reviewer #2: Yes

5. Review Comments to the Author

Reviewer #1: Although this is a topic of interest, it is very difficult to follow and read to medical staffs. This is more of for Information System and Computer Science reader.

For example, baseline characteristics of included patients are not clearly provided in tables.

Definition and how to identify cases with final diagnosis of COVID-19 infection are also not provided.

Severity of COVID-19 infection has also not been demonstrated.

Reviewer #2: This paper presented a new COVID-19 identification method based on the locality-weighted learning and self-organization map (LWL-SOM) strategy for detecting and capturing COVID-19 cases. The topic is interesting, I congrats the authors for your effort. However, some aspects should be improved.

• The abstract can be rewritten to be more meaningful. The authors should add more details about their final results in the abstract. Abstract should clarify what is exactly proposed (the technical contribution) and how the proposed approach is validated.

• Bullet your contribution at the end of the introduction section.

• Many recent papers are missing such as:

&& Deploying machine and deep learning models for efficient data-augmented detection of covid-19 infections. Viruses 12, no. 7 (2020): 769.

%% A Deep Learning Approach to Detect COVID-19 Patients from Chest X-ray Images. AI 1, no. 3 (2020): 418-435.

^^ A Survey on Artificial Intelligence Approaches in Supporting Frontline Workers and Decision Makers for COVID-19 Pandemic. Chaos, Solitons & Fractals (2020): 110337.

etc.

• Figures 1, 5 and 11 not clear, I would ask author to redesign this figure with clear details.

• I recommend adding a Table for related works and show the advantages and disadvantages of each study.

• Authors need to provide justifications for all the parameters setting.

• Overall, the manuscript has potential, if more dataset can be including in the analysis will be beneficial for reaching a concrete conclusion such as:

^^ Combined COVID-19 Dataset, 2020, Mendeley Data, V3, doi: 10.17632/3pxjb8knp7.3

%% Noisy COVID-19 X-ray Dataset, 2020, Mendeley Data, V3, doi: 10.17632/fjg5cbzffh.3

• Please highlight the advantages and disadvantages of your method.

• Do the authors employ any cross-validation scheme? Please, provide details about it.

• I recommend defining all parameters in a Table at the end of Introduction Section.

• Whereas overall English quality is good, a thorough proofreading is required to fix some typos.

6. PLOS authors have the option to publish the peer review history of their article (what does this mean?). If published, this will include your full peer review and any attached files.

Reviewer #1: No

Reviewer #2: **Yes: **Mohamed Hammad

---

## [Author Response · Author response to Decision Letter 0]

17 Nov 2020

Original Manuscript ID: PONE-D-20-32000 

Original Article Title: “SOM-LWL Method for Identification of COVID-19 on Chest X-rays”

To: PLOS ONE Editor

Re: Response to reviewers

Dear Editor,

Thank you for allowing a resubmission of our manuscript, with an opportunity to address the reviewers’ comments.

We are uploading (a) our point-by-point response to the comments (below) (response to reviewers), (b) an updated manuscript with yellow highlighting indicating changes, and (c) a clean updated manuscript without highlights (PDF main document).

Best regards,

Ahmed Hamza Osman

Corresponding Author

Thank you very much for your valuable comments. We have revised the manuscript according to your comments. Your comments help us improve the quality of the paper a lot. Thank you again!

Concern # 1: characteristics of included patients are not clearly provided.

Author response: The characteristics of included patients are updated and reported and now clearer. Please see Section V – dataset subsection B.

Concern # 2: Analysis of recent related papers is missing

Author response: The analysis of recent related papers has been reviewed and reported to be more solid. In addition, the manuscript has been revised accordingly. Please see Section II – Related works.

Concern # 3: figures 1, 5 and 11 are not clear,

Author response: The figures 1, 5 and 11 has been redrawn to be clearer. Please see figures 1, 5 and 11 in the paper.

Concern # 4: it is not clear if cross-validation scheme was used

Author response: The Cross validation scheme has been used and involved in our experiments. The simple idea of holdout by using certain information for testing and the rest for training. In 10-fold cross validation, the contents of one fold are influenced by the contents of other folds. The different cross-validations are applied with independent samples from the COVID-19 dataset to obtain some variations in results and remove any outliers based on averaging. The COVID-19 dataset was divided into 10 folds after the data balancing process. Two different scenarios have been used for the identification and classification of COVID-19 in X-rays. First, the SOM-LWL scheme is trained to classify the X-rays into three classes: COVID-19, Non-COVID-19, and pneumonia. Furthermore, two classes are trained with the SOM-LWL model: the COVID-19 classes and the Non-COVID-19 classes. For triple and binary classification problems, the output of the suggested model is assessed by the 10-fold cross validation process. The training records use 90% of the X-ray images and 10% as the testing stage, the process is performed four times based on the balance of the dataset that has been determined in pre-processing phase. In contrast to the traditional hold-out validation process, this type of validation method provides better results.

Please see Section V – subsection A.

Concern # 5: Proofreading is required to fix typos.

Author response: The English of the paper was revised and the paper was sent to the American Journal Experts (AJE) proofreading services to fix all the typos and the Grammar errors in the paper as well. A proofreading certificate was provided and attached to prove that.

Additional comments after revise version by the Journal Editor:

1) Please ensure that you refer to Figure 3 and Figure 10 in your text as, if accepted, production will need this reference to link the reader to the figure.

The mentioned Figure has been refereed and cited in the text and updated in the list of references. Figure 10 is our own and was produced through the experiments carried out by this study.

Finally, we are very thankful to the anonymous reviewers for his very useful suggestions and comments. I hope the paper could have made an improvement this time.

Yours sincerely

Ahmed Hamza Osman 

Corresponding Author

---

## [Decision Letter · Decision Letter 1]

25 Nov 2020

PONE-D-20-32000R1

SOM-LWL Method for Identification of COVID-19 on Chest X-rays

PLOS ONE

Dear Dr. Osman,

Thank you for submitting your manuscript to PLOS ONE. After careful consideration, we feel that it has merit but does not fully meet PLOS ONE’s publication criteria as it currently stands. Therefore, we invite you to submit a revised version of the manuscript that addresses the points raised during the review process.

Especially, please carefully read the reviewer's comments and address all their concerns.

We look forward to receiving your revised manuscript.

Kind regards,

Yuchen Qiu, Ph.D.

Academic Editor

PLOS ONE

Additional Editor Comments (if provided):

Please carefully read the reviewers' comments and address their concerns.

Especially, the 2nd reviewer mentioned that his concerns are not addressed in this version. I copied his comments in the 1st cycle as follows and please revise your manuscript accordingly.

2nd reviewers comments

This paper presented a new COVID-19 identification method based on the locality-weighted learning and self-organization map (LWL-SOM) strategy for detecting and capturing COVID-19 cases. The topic is interesting, I congrats the authors for your effort. However, some aspects should be improved.

• The abstract can be rewritten to be more meaningful. The authors should add more details about their final results in the abstract. Abstract should clarify what is exactly proposed (the technical contribution) and how the proposed approach is validated.

• Bullet your contribution at the end of the introduction section.

• Many recent papers are missing such as:

&& Deploying machine and deep learning models for efficient data-augmented detection of covid-19 infections. Viruses 12, no. 7 (2020): 769.

%% A Deep Learning Approach to Detect COVID-19 Patients from Chest X-ray Images. AI 1, no. 3 (2020): 418-435.

^^ A Survey on Artificial Intelligence Approaches in Supporting Frontline Workers and Decision Makers for COVID-19 Pandemic. Chaos, Solitons & Fractals (2020): 110337.

etc.

• Figures 1, 5 and 11 not clear, I would ask author to redesign this figure with clear details.

• I recommend adding a Table for related works and show the advantages and disadvantages of each study.

• Authors need to provide justifications for all the parameters setting.

• Overall, the manuscript has potential, if more dataset can be including in the analysis will be beneficial for reaching a concrete conclusion such as:

^^ Combined COVID-19 Dataset, 2020, Mendeley Data, V3, doi: 10.17632/3pxjb8knp7.3

%% Noisy COVID-19 X-ray Dataset, 2020, Mendeley Data, V3, doi: 10.17632/fjg5cbzffh.3

• Please highlight the advantages and disadvantages of your method.

• Do the authors employ any cross-validation scheme? Please, provide details about it.

• I recommend defining all parameters in a Table at the end of Introduction Section.

• Whereas overall English quality is good, a thorough proofreading is required to fix some typos.

Reviewers' comments:

Reviewer's Responses to Questions

**Comments to the Author**

1. If the authors have adequately addressed your comments raised in a previous round of review and you feel that this manuscript is now acceptable for publication, you may indicate that here to bypass the “Comments to the Author” section, enter your conflict of interest statement in the “Confidential to Editor” section, and submit your "Accept" recommendation.

Reviewer #1: All comments have been addressed

Reviewer #2: (No Response)

2. Is the manuscript technically sound, and do the data support the conclusions?

Reviewer #1: Partly

Reviewer #2: Partly

3. Has the statistical analysis been performed appropriately and rigorously? 

Reviewer #1: Yes

Reviewer #2: No

4. Have the authors made all data underlying the findings in their manuscript fully available?

Reviewer #1: Yes

Reviewer #2: Yes

5. Is the manuscript presented in an intelligible fashion and written in standard English?

Reviewer #1: (No Response)

Reviewer #2: Yes

6. Review Comments to the Author

Reviewer #1: - Abstracts should include more details on the AUC of each machine learning model vs conventional regression models.

-Limitations of AI.

-I suggest that there be at least a paragraph on the limitations of research using AI.

-I have already mentioned that the approach used might not be the best one and thus deciding which one to use is important. -AI can be an exercise if mining for P values. That is dependent on the data that the computer has to analyze. One problem with machine learning such as CNNs is that the filters used by the computer are not clearly identifiable to the researchers. -- The computer could be reading the bar code on the CT scan for all we know. Thus, understanding biological mechanisms is limited in AI and understanding the computer’s methods can be limited. You basically get a result in many instances. This should be mentioned.

Reviewer #2: All my concerns not addressed. So, I recommend not accepting this paper in this version. I recommend another revision as I'm not satisfied with this version.

7. PLOS authors have the option to publish the peer review history of their article (what does this mean?). If published, this will include your full peer review and any attached files.

Reviewer #1: No

Reviewer #2: No

---

## [Author Response · Author response to Decision Letter 1]

14 Dec 2020

Dear Editor,

Thank you for allowing a re-submission of our manuscript, with an opportunity to address the reviewers’ comments.

We are uploading (a) our point-by-point response to the comments (below) (response to reviewers), (b) an updated manuscript with yellow highlighting indicating changes, and (c) a clean updated manuscript without highlights.

---

## [Decision Letter · Decision Letter 2]

11 Jan 2021

PONE-D-20-32000R2

SOM-LWL Method for Identification of COVID-19 on Chest X-rays

PLOS ONE

Dear Dr. Osman,

Thank you for submitting your manuscript to PLOS ONE. After careful consideration, we feel that it has merit but does not fully meet PLOS ONE’s publication criteria as it currently stands. Therefore, we invite you to submit a revised version of the manuscript that addresses the points raised during the review process.

We look forward to receiving your revised manuscript.

Kind regards,

Yuchen Qiu, Ph.D.

Academic Editor

PLOS ONE

Reviewers' comments:

Reviewer's Responses to Questions

**Comments to the Author**

1. If the authors have adequately addressed your comments raised in a previous round of review and you feel that this manuscript is now acceptable for publication, you may indicate that here to bypass the “Comments to the Author” section, enter your conflict of interest statement in the “Confidential to Editor” section, and submit your "Accept" recommendation.

Reviewer #1: All comments have been addressed

Reviewer #2: All comments have been addressed

2. Is the manuscript technically sound, and do the data support the conclusions?

Reviewer #1: Yes

Reviewer #2: Yes

3. Has the statistical analysis been performed appropriately and rigorously? 

Reviewer #1: Yes

Reviewer #2: Yes

4. Have the authors made all data underlying the findings in their manuscript fully available?

Reviewer #1: Yes

Reviewer #2: Yes

5. Is the manuscript presented in an intelligible fashion and written in standard English?

Reviewer #1: Yes

Reviewer #2: Yes

6. Review Comments to the Author

Reviewer #1: I have no competing interests. The authors have responded appropriately. There are limitations to the study but they are stated.

Reviewer #2: The authors have addressed almost of reviewer's concerns and the revised version of the manuscript appears to be good.

Minor revision is needed:

1. Please bullet your contribution at the end of the introduction Section.

2. Table I, please add the used methods and the performance for each reference.

3. Add references for your future work (e.g. give references for the data).

4. Abstract, still need to add more details about your final results.

7. PLOS authors have the option to publish the peer review history of their article (what does this mean?). If published, this will include your full peer review and any attached files.

Reviewer #1: No

Reviewer #2: **Yes: **Mohamed Hammad

---

## [Author Response · Author response to Decision Letter 2]

15 Jan 2021

Dear Editor,

Thank you for allowing a resubmission of our manuscript, with an opportunity to address the reviewers’ comments.

We are uploading (a) our point-by-point response to the comments (below) (response to reviewers), (b) an updated manuscript with yellow highlighting indicating changes, and (c) a clean updated manuscript without highlights.

Best regards,

Ahmed Hamza Osman

Corresponding Author

Reviewer # 1

I have no competing interests. The authors have responded appropriately. 

Thank you very much for your valuable comments. We have revised the manuscript according to your comments. Your comments help us improve the quality of the paper a lot. Thank you again!

Reviewer # 2

The authors have addressed almost of reviewer's concerns and the revised version of the manuscript appears to be good.

Minor revision is needed:

Concern # 1: Please bullet your contribution at the end of the introduction Section.

Author response: 

The paper contribution was added at the end of the introduction section to be clearer according to the reviewer comment. Please see the end of the introduction section in the manuscript.

Concern # 2: Table I, please add the used methods and the performance for each reference.

Author response: Table I was revised according to the reviewer comments by adding the used methods/techniques and its performances. Please see Table I in the manuscript.

Concern # 3: Add references for your future work (e.g. give references for the data).

Author response: The future work section has been revised by adding some references that will be improved this research in future. Please see the future work section.

Concern # 4: Abstract, still need to add more details about your final results. 

Author response: The abstract has been revised and the finding of final results has been reported and highlighted too. Please see the abstract of the paper.

Finally, we are very thankful to the anonymous reviewers for his very useful suggestions and comments. I hope the paper could have made an improvement this time.

Yours sincerely

Ahmed Hamza Osman 

Corresponding Author

---

## [Decision Letter · Decision Letter 3]

3 Feb 2021

SOM-LWL Method for Identification of COVID-19 on Chest X-rays

PONE-D-20-32000R3

Dear Dr. Osman,

We’re pleased to inform you that your manuscript has been judged scientifically suitable for publication and will be formally accepted for publication once it meets all outstanding technical requirements. Meanwhile, you may need to revise the citation as suggested by reviewers, and submit a new version for publication.

Kind regards,

Yuchen Qiu, Ph.D.

Academic Editor

PLOS ONE

Additional Editor Comments (optional):

Please revise the citation as suggested by the reviewer.

Reviewers' comments:

Reviewer's Responses to Questions

**Comments to the Author**

1. If the authors have adequately addressed your comments raised in a previous round of review and you feel that this manuscript is now acceptable for publication, you may indicate that here to bypass the “Comments to the Author” section, enter your conflict of interest statement in the “Confidential to Editor” section, and submit your "Accept" recommendation.

Reviewer #2: All comments have been addressed

2. Is the manuscript technically sound, and do the data support the conclusions?

Reviewer #2: Yes

3. Has the statistical analysis been performed appropriately and rigorously? 

Reviewer #2: Yes

4. Have the authors made all data underlying the findings in their manuscript fully available?

Reviewer #2: Yes

5. Is the manuscript presented in an intelligible fashion and written in standard English?

Reviewer #2: Yes

6. Review Comments to the Author

Reviewer #2: I appreciate the authors’ efforts to revise the manuscript. In its current form, the manuscript addresses the major concerns I have raised.

My only minor comment:

Authors cited many references for AI applications in different fields from ref 29-41, I think it is too much, you should cite only one reference for each application. Also, there are some recent papers related to COVID-19 not cited that I recommend citing them such as:

&& Efficient deep learning approach for augmented detection of Coronavirus disease. Neural Computing and Applications (2021): 1-18.

^^ COVIDScreen: Explainable deep learning framework for differential diagnosis of COVID-19 using chest X-Rays. Neural Computing and Applications (2021): 1-22.

$ An Efficient Method for Coronavirus Detection Through X-rays using deep Neural Network. Current medical imaging (2021).

## Evaluation of deep learning-based approaches for COVID-19 classification based on chest X-ray images. Signal, Image and Video Processing (2021): 1-8.

** COVINet: a convolutional neural network approach for predicting COVID-19 from chest X-ray images. Journal of Ambient Intelligence and Humanized Computing (2021): 1-13.

7. PLOS authors have the option to publish the peer review history of their article (what does this mean?). If published, this will include your full peer review and any attached files.

Reviewer #2: **Yes: **Mohamed Hammad

---

## [Editor Report · Acceptance letter]

10 Feb 2021

PONE-D-20-32000R3 

SOM-LWL Method for Identification of COVID-19 on Chest X-rays 

Dear Dr. Osman:

I'm pleased to inform you that your manuscript has been deemed suitable for publication in PLOS ONE. Congratulations! Your manuscript is now with our production department. 

Kind regards, 

on behalf of

Dr. Yuchen Qiu 

Academic Editor

PLOS ONE